

# Optimizing agricultural data security: harnessing IoT and AI with Latency Aware Accuracy Index (LAAI)

Omar Bin Samin[1,2], Nasir Ahmed Abdulkhader Algeelani[2,3], Ammar Bathich[2], Maryam Omar[4], Musadaq Mansoor[5] and Amir Khan[1]

[1] Center of Excellence in IT, Institute of Management Sciences (IMSciences), Peshawar, Peshawar, Pakistan
[2] Faculty of Computer & Information Technology, Al-Madinah International University, Kuala Lumpur, Malaysia
[3] Institute of Engineering, Electrical and Electronic Engineering, Hanze University of Applied Science, Groningen, Netherlands
[4] Veritas Analytica, Leverify, Seattle, United States of America
[5] Faculty of Computer Science and Engineering, Ghulam Ishaq Khan Institute of Engineering Sciences and Technology, Topi, Pakistan

## ABSTRACT

The integration of Internet of Things (IoT) and artificial intelligence (AI) technologies into modern agriculture has profound implications on data collection, management, and decision-making processes. However, ensuring the security of agricultural data has consistently posed a significant challenge. This study presents a novel evaluation metric titled Latency Aware Accuracy Index (LAAI) for the purpose of optimizing data security in the agricultural sector. The LAAI uses the combined capacities of the IoT and AI in addition to the latency aspect. The use of IoT tools for data collection and AI algorithms for analysis makes farming operation more productive. The LAAI metric is a more holistic way to determine data accuracy while considering latency limitations. This ensures that farmers and other end-users are fed trustworthy information in a timely manner. This unified measure not only makes the data more secure but gives farmers the information that helps them to make smart decisions and, thus, drives healthier farming and food security.

## INTRODUCTION

In the ever-evolving landscape of agriculture, it has become crucial to incorporate state-of-the-art technologies in order to enhance production, optimize resource utilization, increase efficiency, and guarantee the implementation of sustainable practices. The digitization in agricultural sectors is experiencing rapid exponential growth. Digitization is the term used to describe the procedure by which data acquired physically (*e.g.*, from sensors) is converted into a format that can be read by a computer (*Mondejar et al., 2021*). This rate of digitization in the agricultural sector leads to an exponential increase in the quantity of sensitive data that is generated, processed, and stored. The increase in the volume of data emphasizes the evident need for comprehensive safety measures in order to protect

Corresponding author
Omar Bin Samin,
omar.samin@imsciences.edu.pk

sensitive agricultural data linked to soil conditions, weather patterns, and agricultural productivity, *etc*. However, due to computational constraints of existing systems, they lack the capacity to effectively analyze vast amounts of data within a limited timeframe. As a result, they become more susceptible to a range of attacks, including Distributed Denial of Service (DDoS), SQL attack, MITM attack, backdoor attack, port scanning, and ransomware attack, among others (*Tomer & Sharma, 2022*; *Ramadan, 2022*).

Agriculture is vital for the survival, expansion, and progress of the human race as it supplies the vast majority of their food (*Timmis & Ramos, 2021*). It is a critical sector that significantly influences and sustains the economy of a country (*Adekoya et al., 2022*). Agriculture exhibits an extensive effect on the economy, incorporating various dimensions including the generation of income, employment, foreign exchange earnings, and food security. The convergence of technology and agriculture has led to the emergence of a paradigm in which artificial intelligence (AI) and the Internet of Things (IoT) are utilized to strengthen agricultural defenses against potential cyber threats (*Seng, Ang & Ngharamike, 2022*). By utilizing this collaborative strategy, stakeholders are not only empowered to make well-informed decisions, but the integrity of agricultural data is also ensured, which contributes to the development of a more sustainable and resilient agricultural ecosystem.

Despite considerable effort invested in agricultural IoT security, a significant amount of prior research is based on either a restricted dataset or a relatively narrow spectrum of IoT attacks. Furthermore, the latency component of the system has been neglected in the vast majority of prior studies. The term latency refers to the potential delay (*Lakhan et al., 2021*) that can occur during data processing or task execution (*Kishor, Chakraborty & Jeberson, 2021*). It quantifies the time interval between the initiation and completion of the process. This study proposes a state-of-the-art evaluation metric, Latency Aware Accuracy Index (LAAI), capable of identifying the most suitable AI subdomain model incorporating latency factor for classifying and detecting various types of malicious attacks in the agricultural IoT domain utilizing EdgeIIoTset. The proposed research utilizes machine learning (ML) models (*i.e.,* naïve Bayes and Decision Tree) and deep learning (DL) model (*i.e.,* artificial neural network) for IoT traffic multi-class classification. Therefore, the objective is to connect computationally intensive artificial intelligence methods with resource limited agricultural IoT devices in order to enhance the security and efficiency of the system, while also taking system's latency into consideration.

The organizational specifics of the article are as follows. 'Literature Review', examines previous research on detection and classification of malicious IoT activities using ML and DL models. In 'Methodology', the implementation specifics are discussed in further depth, and in 'Results & Discussion', the study's key findings are presented. Finally, 'Conclusion', concludes the paper by specifying the most suitable AI model for the EdgeIIoTset, based on the evaluations in the study.

## LITERATURE REVIEW

This section examines previous research on the security of IoT data and networks, with a particular focus on the identification and categorization of harmful IoT network activity.

*Fu et al. (2023)* proposed an approach aiming to secure agricultural information systems. The study proposed a Double Deep Q-Network (DDQN) algorithm utilizing the geography position information to quickly optimize UAV deployment positions without complicated derivations and improve security-checking efficiency in agriculture environments. In addition to that, the inclusion of convolutional neural network-long short-term memory (CNN-LSTM) lead towards developing a new pattern for intrusion detection system not only in IoT but also for other fields as well. Thus, by controlling the data transmission and network structure construction using long short-term memory (LSTM) and convolutional neural network (CNN) respectively security of precision agriculture was ensured. In order to evaluate the performance of the algorithms under a variety of parameter configurations, the experiments were complemented with simulation that further inform how to deploy UAVs efficiently. The IoT intrusion detection system based on the CNN-LSTM algorithm achieved an accuracy of 93.5% and a detection rate of 94.4% utilizing the KDD-CUP99 dataset.

*Aldhyani & Alkahtani (2023)* utilized DL models to reduce the escalating challenge of DDoS attacks on Agriculture 4.0 networks. The research presented an adaptive intrusion detection system (IDS) that used architectures like LSTM, combination of CNN and LSTMs CNN–LSTM for DDoS detection. To develop and evaluate the developed system, the study used the CIC-DDoS2019 dataset which has been generated from a network monitored by CRCFlowMeter-V3. The authors used frequently studied standard network traffic datasets (NetBIOS, Portmap, Syn, UDPLag and benign packets) to train and evaluation the model. Performance metrics such as precision, recall, F1-score and accuracy highlight the high-performance capabilities of CNN–LSTM model and achieved both almost 100% accuracy score.

*Javeed et al. (2023)* presented an IDS for the edge-based smart agriculture in extreme conditions. Using the CIC-IDS2018, ToN-IoT, and Edge-IIoT datasets, they incorporated edge computing for providing real-time intrusion detection. The objective of the study was to enhance security in smart agricultural systems, which is critical for ensuring that they remain operationally resilient in adverse environmental settings. The necessity of intrusion detection to safeguard agricultural infrastructure from cyber threats, which was emphasized by the authors and their advanced algorithms combined with edge computing capabilities The accuracy rate of their system also showed satisfaction with a record between 99.51% and 99.91%, which proves the importance of this model as an extra layer protecting security in smart agriculture.

*Saha et al. (2021)* conducted a study focused on the security aspect of smart agriculture. The authors delved into IoT applications such as real time crop monitoring, precision farming and data analytics by using sensors for soil moisture estimation and weather monitoring. Despite technological advancements, challenges such as software simplicity and secure data transmission were noted. Seamless integration with agriculture, skilled workforce, low power sensors were some of the matters that they brought out in their study. They also emphasized better connectivity, remote management, and enhanced security measures. The team suggested that future research should focus on the importance of security in IoT enabled agricultural systems for uninterrupted services.

*Setiadi, Kesiman & Aryanto (2021)* described a way to find Denial-of-Service (DoS) attacks on an IoT platform using the NSL-KDD and KDD'99 datasets and the naïve Bayes method. DoS attacks are very dangerous to computer networks because they break down PCs or servers that are linked to the internet. The main goal of intrusion detection study is to find the best way to choose features so that IoT attacks and intrusions do not hurt the effectiveness of IDS. The study produced results with a 99% recall rate, 50% precision, and 64% accuracy rate.

*Manimurugan (2021)* developed an IoT-Fog-Cloud model using the UNSW-NB dataset, incorporating PCA and improving naïve Bayes for anomaly detection. This study explored the feasibility of integrating cloud and fog computing with the IoT, laying the foundation for securing IoT-enabled smart city applications. Fog computing provides a wide range of services as it provides IoT services, while cloud computing provides storage management. This work uses PCA-based improved naïve Bayes (INB) method to detect anomalies in network based intrusion detection systems (NIDS). INB-PCA performed well on the UNSW-NB15 data set, achieving 92.48% accuracy and 95.35% detection rates.

*Douiba et al. (2023)* proposed to monitor IoT systems for inconsistencies. This study used gradient boosting (GB) and CatBoost to improve IoT security IDS models. The model scored well on the NSL-KDD, BoT-IoT, IoT-23 and Edge-IIoTset datasets. It achieved about 99.9% accuracy, recall and accuracy in record discovery and computation time. This anomaly-based IDS contributes significantly to the enhanced security of the Internet of Things, especially when combined with dark computing costs. The paper emphasizes the effectiveness of the model and credits GPU usage, gradient boosting, and CatBoost for its performance.

*Guezzaz et al. (2021)* proposed an approach for network intrusion detection utilizing Decision Tree with improved data quality, employing the NSL-KDD and CICIDS2017 datasets. Through preprocessing and entropy Decision Tree based feature selection, the method boosted detection rates and data quality, validated on the NSL-KDD and CICIDS2017 datasets. This approach showcased advantages and heightened accuracy, setting the stage for future integration of additional machine learning techniques. The suggested model, demonstrated an accuracy rate of 99% for NSL-KDD, while the evaluation on the CICIDS2017 dataset attests to an accuracy level of 98%.

*Jamal, Hayat & Nasir (2022)* proposed a method for the identification and classification of malware in IoT networks by utilizing artificial neural networks with the Ton_IoT dataset. This research investigated the efficacy of neural networks in identifying and classifying malware inside an IoT network. The utilized Ton_IoT dataset consisting of 461,043 records, with 300,000 benign occurrences and 161,043 malicious incidents. Based on the traffic on the IoT network, the proposed method achieves an accuracy of 94.17% in detecting malware and 97.08% in classifying classes of malware families.

*Al-Zewairi, Almajali & Ayyash (2020)* used shallow and deep artificial neural network classifiers to identify undetected security breaches. To find the best datasets for unusual attacks, the authors examined and standardized novel attack classification. They evaluated new anomaly-based intrusion detection techniques for detecting unknown threats using two datasets and artificial neural network models for binary and multi-class classification.

The study claimed to have produced encouraging results. The malware detection accuracy was 94.17%, and the malware family classification accuracy was 97.08%.

*Gopi et al. (2021)* proposed refining an artificial neural network based model for identifying DDoS assaults on multimedia IoT devices. An artificial neural network (ANN) learns to recognize feature vector patterns and predict output. This is critical for multimodal IoT DDoS attack detection. Training minimizes the loss function by employing one dimensional search spaces and mathematical procedures. The system classifies packets as normal or malicious by collecting signatures and behavior patterns using the IDS. The model uses the ML training technique and data dimension reduction to detect DDoS attempts, and it separates compromised packets incredibly well to protect network capacity and utilization.

*Osman et al. (2021)* developed an ANN model to detect instances of lower-rank attacks in RPL-based IoT networks. After pre-processing a dataset of RPL protocol control messages with a Random Forest classifier to discover the most essential features, an ANN model is trained to categorize messages as benign or malignant. The model is having an input layer, including one or more hidden layers, and the layer of output, with each layer consisting of many neurons. The model is trained using a specific dataset and then evaluated using a different dataset. The suggested model is assessed in comparison to other machine learning methods and shows positive results in detecting reduced rank attacks in RPL-based IoT networks. The article claims an accuracy rate of 97.01%.

The literature mentioned above regarding IoT security pertains to either a restricted range of IoT attacks or a meager quantity of data. Additionally, almost all of previous research overlooked the latency aspect of the system. Furthermore, comparing and ranking the performance of different ML and DL classifiers using diverse datasets is not a reasonable approach. The proposed work will tackle all of these concerns. This research aims to determine the best AI model for detecting various IoT attack classes in agricultural applications using different data volumes from the same dataset, while also considering latency factor crucial for optimizing the performance and security of IoT systems.

## METHODOLOGY

At present, various ML and DL methods specifically developed for identifying and categorizing both normal and malicious data traffic in agricultural IoT are claiming their effectiveness. Nevertheless, evaluating their performance and efficacy in a wide range of scenarios, which include different datasets with varying amounts of training and testing records, as well as various types of IoT attacks, continues to be a challenging task. Furthermore, while latency is a crucial metric for assessing the performance of IoT systems, prior studies has predominantly overlooked its impact on performance evaluation. To address this gap and support these claims, the proposed research aims to lay a robust groundwork for the optimal selection of AI model incorporating latency factors. This will be accomplished by comparing the naïve Bayes, Decision Tree, and ANN models on a specific dataset designed exclusively for agricultural applications in the IoT domain. These three models, have been purposefully chosen for their application to the diversified and

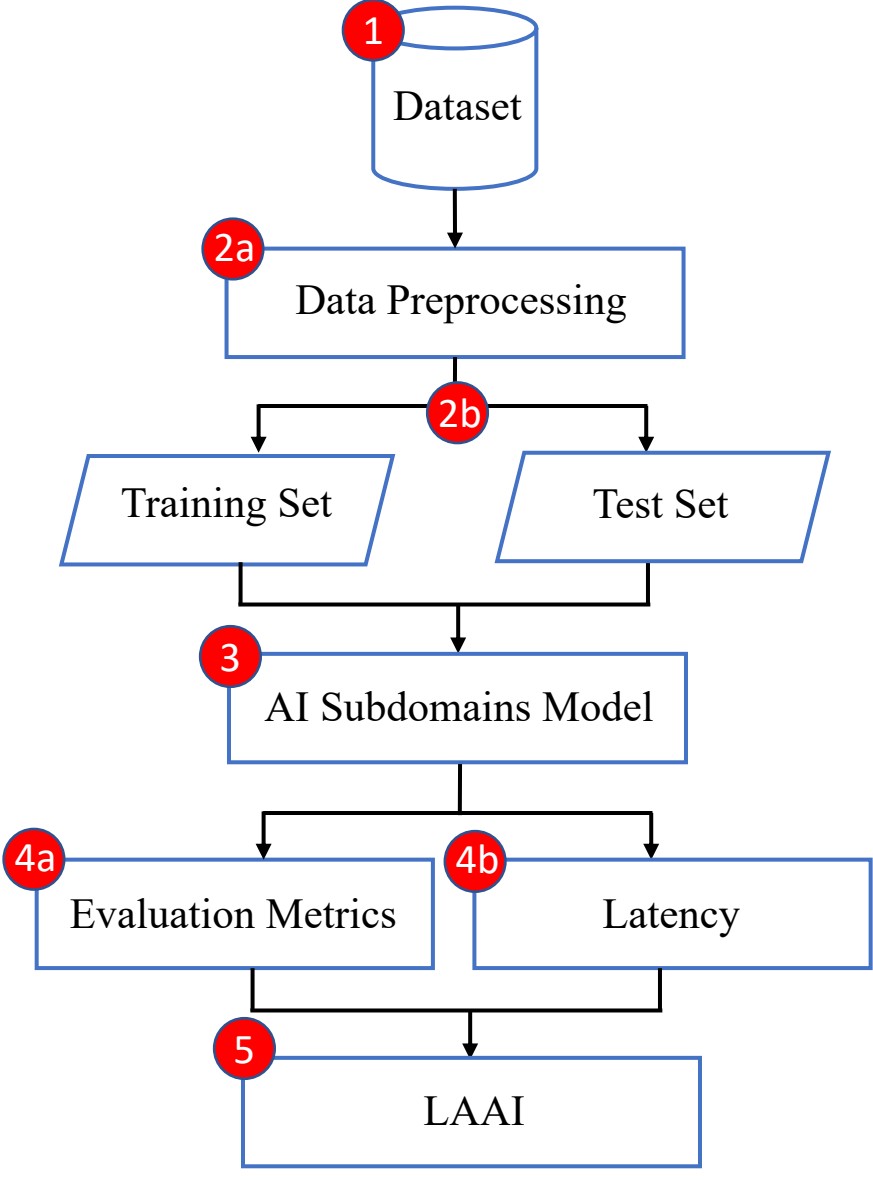

**Figure 1  Intended methodology.**

cutting-edge dataset known as the Edge-IIoTset which is publicly available for research purposes. The objective of this research is to contribute to the understanding of AI domain by introducing a cutting-edge evaluation metric called the Latency Aware Accuracy Index (LAAI) that can determine the most appropriate AI subdomain model through integrating the achieved accuracy and latency factor. This will enable prompt and essential responses to be taken. The intent of this study is to establish a foundational reference that will indicate the most effective model for classifying agricultural IoT traffic within the specified dataset. The Fig. 1 represents the proposed pipeline of an intended methodology.

**Table 1 ML-Edge-IIoTset details (*Ferrag et al., 2022*).**

| S# | Class | Data records |
| --- | --- | --- |
| 1 | Normal_IoT_Data | 24,301 |
| 2 | Backdoor_Attack | 10,195 |
| 3 | DDoS_HTTP_Attack | 10,561 |
| 4 | DDoS_ICMP_Attack | 14,090 |
| 5 | DDoS_TCP_Attack | 10,247 |
| 6 | DDoS_UDP_Attack | 14,498 |
| 7 | Fingerprinting_Attack | 1,001 |
| 8 | MITM_Attack | 1,214 |
| 9 | Password_Attack | 9,989 |
| 10 | Port_Scanning_Attack | 10,071 |
| 11 | Ransomware_Attack | 10,925 |
| 12 | SQL_Injection_Attack | 10,311 |
| 13 | Uploading_Attack | 10,269 |
| 14 | Vulnerability_Scanner_Attack | 10,076 |
| 15 | Cross-site_Scripting_(XSS)_Attack | 10,052 |
| | Total records | 157,800 |

## Dataset

In the context of the agricultural IoT, the identification and classification of intrusions and attacks are significantly influenced by the selection of an appropriate dataset.

The EdgeIIoTset (*Ferrag et al., 2022*), a cutting-edge public IoT security dataset, was intentionally selected for this particular objective. The dataset contains extensive statistics on IoT traffic, consisting of a total of 15 categories. The classes consist of 14 categories that represent different sorts of IoT attacks, along with one class that represents regular IoT data. In the proposed investigation, the utilization of this public dataset is paramount, as public datasets provide a valuable platform for researchers to compare and benchmark their contributions against the work of others in the field. The dataset is obtained from sensors that measure variables including temperature, humidity and water level, among other things, and is suitable for IoT security application in agriculture. Edge-IIoTset is composed of two subsets, namely ML-Edge-IIoTset and DNN-Edge-IIoTset, provided by the developers of the dataset (see Tables 1 and 2).

The Edge-IIoTset dataset encompasses diverse IoT data traffic classifications, which are outlined as follows:

1. **Normal_IoT_Data:** Authentic data and legitimate requests.
2. **Backdoor_Attack:** Installs backdoors in IoT networks by taking control of the vulnerable system components.
3. **DDoS_HTTP_Attack:** Executes the HTTP manipulation and sending the spam queries.
4. **DDoS_ICMP_Attack:** Bombards the target device with a disproportionate amount of Internet Control Message Protocol (ICMP) echo requests, commonly called pings.
5. **DDoS_TCP_Attack:** Overloads the target device with a disproportionate amount of SYN requests to ensure it is unable to accept any new connections.

**Table 2  DNN-Edge-IIoTset details (*Ferrag et al., 2022*).**

| S# | Class | Data records |
|---|---|---|
| 1 | Normal_IoT_Data | 1,615,643 |
| 2 | Backdoor_Attack | 24,862 |
| 3 | DDoS_HTTP_Attack | 49,911 |
| 4 | DDoS_ICMP_Attack | 116,436 |
| 5 | DDoS_TCP_Attack | 50,062 |
| 6 | DDoS_UDP_Attack | 121,568 |
| 7 | Fingerprinting_Attack | 1,001 |
| 8 | MITM_Attack | 1,214 |
| 9 | Password_Attack | 50,153 |
| 10 | Port_Scanning_Attack | 22,564 |
| 11 | Ransomware_Attack | 10,925 |
| 12 | SQL_Injection_Attack | 51,203 |
| 13 | Uploading_Attack | 37,634 |
| 14 | Vulnerability_Scanner_Attack | 50,110 |
| 15 | Cross-site_Scripting_(XSS)_Attack | 15,915 |
| | Total records | 2,219,201 |

6. **DDoS_UDP_Attack:** Overwhelms the target device with a vast amount of User Datagram Protocol (UDP) packets to disrupt the processing and responses.
7. **Fingerprinting_Attack:** Analyze IoT data packets in examining IoT devices and servers.
8. **MITM_Attack:** Eavesdrops on the communication between two IoT devices or between the IoT device and the server.
9. **Password_Attack:** Decodes the password of an IoT device to gain illegal entry.
10. **Port_Scanning_Attack:** Aids hackers in determining vulnerabilities inside the IoT network.
11. **Ransomware_Attack:** Encrypts the IoT data or systems to disable access until a ransom is paid.
12. **SQL_Injection_Attack:** Utilizes the SQL queries to read, insert, update and delete the sensitive information from the database.
13. **Uploading_Attack:** Uploads the malware infected file that spreads infectious command, thus takes control of the device.
14. **Vulnerability_Scanner_Attack:** Scans and notifies the security flaws present inside the IoT network.
15. **Cross-site_Scripting_(XSS)_Attack:** Compromises the confidential information through deploying the vicious scripts to the victims.

It is recommended by the Edge-IIoTset developers that ML-Edge-IIoTset be utilized when applying ML techniques, whereas DNN-Edge-IIoTset be utilized when applying deep learning techniques. In order to meet the needs of investigators, each of these subsets comprises varying quantities of data while still encompassing all classes from the entire dataset. This study employs both ML-EdgeIIoTset and DNN-EdgeIIoTset to evaluate the performance of three selected classifiers/ models (*i.e.,* naïve Bayes, Decision Tree, and

ANN). The objective is to investigate the impact of varying data quantities on classifier's performance and latency.

## Data preprocessing

The Edge-IIoTset comprises 63 features in total. A total of 20 out of 63 are classified as object type, and 17 of them have Null values, zero values, or mixed data types. The object attributes consist of integer, floating-point, and string data labeled as Not a Number (NaN), which hold no importance in the application of machine learning and deep learning models. The datasets ML-Edge-IIoTset and DNN-Edge-IIoTset have been preprocessed by eliminating attributes with only zero, null, and NaN values, resulting in 46 attributes available for classification. The datasets exhibit a notable degree of imbalance, as illustrated in Tables 1 and 2. An imbalance in the distribution of classes within the training dataset is the root cause of the imbalanced classification issue. A data augmentation technique known as Synthetic Minority Oversampling Technique (SMOTE) is employed to balance these unbalanced datasets, thereby resolving the imbalance issue and producing balanced datasets. In order to achieve classification parity, SMOTE produces synthetic samples that represent the minority class. This is achieved through the process of arbitrarily selecting instances from the minority class, identifying their k-nearest neighbors (which are frequently members of the same class), and generating synthetic examples by combining the selected instance with its neighbors in a linear fashion. This process is repeated iteratively until the desired degree of class equilibrium is achieved. An equal portion of both datasets, *i.e.,* 70% is allocated for training and the remaining 30% is designated for testing.

## AI subdomains model

The proposed study settled on naïve Bayes (*Nababan, 2024*; *Listiyono et al., 2024*; *Zhang, Jiang & Webb, 2023*), Decision Tree (*Zhuang et al., 2024*; *Coscia et al., 2024*; *Samin et al., 2023*), and ANN (*Mustaqim, Fadil & Tyas, 2023*; *Muñoz-Zavala et al., 2024*; *Yaman, Karakaya & Köküm, 2024*) as the selected methods. Naïve Bayes and Decision Tree are categorized as machine learning models, whereas ANN is classified as a deep learning model. Each of these models will be employed for IoT traffic multi-class classification.

## Evaluation metrics

Evaluation metrics are used to measure the efficiency of statistics, machine learning, and deep learning models. Four assessment metrics, namely accuracy (*Gao et al., 2021*; *Sharma & Guleria, 2023*), precision (*Fränti & Mariescu-Istodor, 2023*; *Mulla & Gharpure, 2023*), recall (*Fränti & Mariescu-Istodor, 2023*; *Alkaissy et al., 2023*), and F1-score (*Obi, 2023*; *Fourure et al., 2021*), are used to evaluate and compare the performances of naïve Bayes, Decision Tree and ANN on both sub datasets (*i.e.,* ML-Edge-IIoTset and DNN-Edge-IIoTset). The mathematical formulas for all employed evaluation metrics are as follows:

$$Accuracy = \frac{T_p + T_n}{T_p + T_n + F_p + F_n}$$

$$Precision = \frac{T_p}{T_p + F_p}$$

$$Recall = \frac{T_p}{T_p + F_n}$$

$$F_1 - Score = 2 * \frac{Precission * Recall}{Precission + Recall}$$

where, $T_p$ represents true positive, $T_n$ represents true negative, $F_p$ represents false positive and $F_n$ represents false negative. An instance of $T_p$ is an outcome in which the model accurately anticipates the positive class. A $T_n$ is an equivalent outcome in which the model accurately anticipates the negative class. An incorrect positive class prediction by the model is denoted by $F_p$ outcome, while an incorrect negative class prediction is denoted by $F_n$ outcome.

### Latency

In the field of computing, latency refers to the potential delay (*Lakhan et al., 2021*) that can occur during data processing or task execution (*Kishor, Chakraborty & Jeberson, 2021*). It quantifies the time interval between the initiation and completion of the process. This indicator has been developed specifically to quantify the level of responsiveness exhibited by the system (*Khayat et al., 2023*). The latency can be calculated as:

$$Latency = \frac{Total\ Time}{Number\ of\ Predictions}$$

where, *Total Time* is the sum duration of the system in analyzing a certain number of predictions and *Number of Predictions* is the population of the predictions.

### LAAI

Although, the current evaluation measures adequately address the specific objectives, however, they fail to prioritize a key element, latency during the evaluation of IoT devices. The proposed LAAI is a novel and comprehensive evaluation measure that incorporates accuracy and latency in evaluating the performance of a model in a practical application and real world scenario. The LAAI is expressed as:

$$LAAI = \frac{Accuracy}{1 + Latency}$$

where, *Accuracy* is the ratio of the correct predictions to the total number of predictions made about the system (*Gao et al., 2021*; *Sharma & Guleria, 2023*) and *Latency* quantifies the potential input–output delay experienced during processing (*Kishor, Chakraborty & Jeberson, 2021*). To normalize the results, *Accuracy* is divided by *1 + Latency*. *LAAI* is the hence normalized term that represents a balance between latency and accuracy. Higher LAAI scores indicate that the system operates more efficiently in terms of both accuracy and latency. Conversely, a low LAAI score means that the model might not be processing the prediction data quickly enough, which is not viable in scenarios that require a fast response, despite the high accuracy of the system.

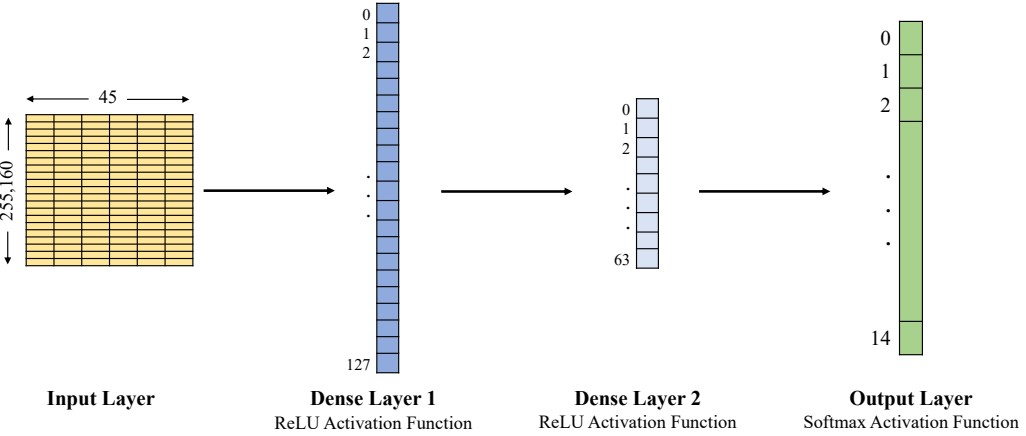

**Figure 2** ANN architecture used with ML-EdgeIIoT-set.

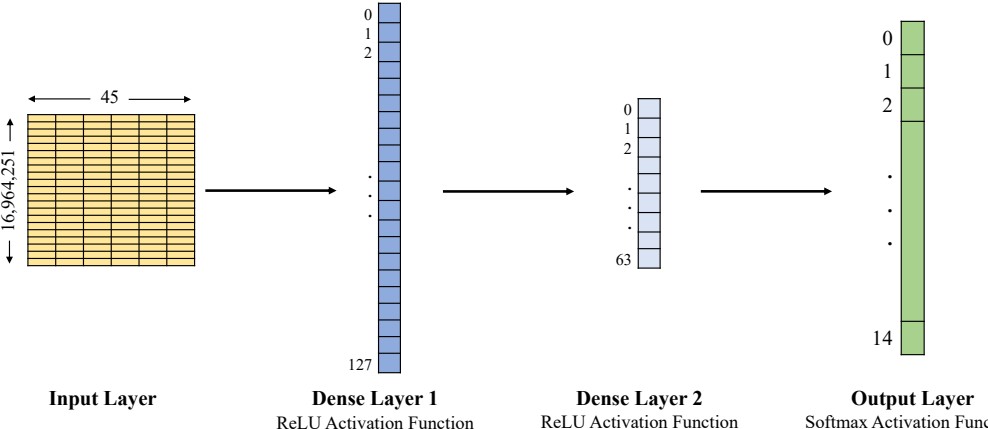

**Figure 3** ANN architecture used with DNN-EdgeIIoT-set.

## RESULTS & DISCUSSION

In this study, we examined the Edge-IIoTset open-access dataset of IoT data traffic to build the training and testing datasets. This study compares three approaches: naïve Bayes, Decision Tree and ANN, in order to classify agricultural IoT data traffic into benign and malicious categories using the selected dataset. The Decision Tree has been set up with a maximum depth of 7, and the structure of the ANN is expressed by the whole 4 layers, 1 input layer, 2 dense layers and 1 output layer (see Figs. 2 and 3). Evaluation metrics including accuracy, precision, recall, F1-score, and latency are used to forecast and evaluate the performance of the three selected models for the selected datasets (see Table 3 for the detailed results).

In Figs. 2 and 3, the first layer is an input layer that represents input data (*i.e.,* 255,160 × 45 for ML-EdgeIIoT-set and 16,964,251 × 45 for DNN-EdgeIIoT-set) to the model. The second layer is a fully connected layer abbreviated as Dense Layer 1 and consists

**Table 3  Evaluation metrics with latency.**

| AI model | Dataset | Accuracy | Precision | Recall | F1-Score | Latency (ms) |
|---|---|---|---|---|---|---|
| Naïve Bayes | ML-Edge-IIoTset | 0.460 | 0.490 | 0.462 | 0.434 | 0.00030 |
| Decision tree | ML-Edge-IIoTset | 0.720 | 0.681 | 0.720 | 0.682 | 0.00011 |
| ANN | ML-Edge-IIoTset | 0.975 | 0.976 | 0.975 | 0.975 | 0.00765 |
| Naïve Bayes | DNN-Edge-IIoTset | 0.450 | 0.467 | 0.448 | 0.410 | 0.00046 |
| Decision tree | DNN-Edge-IIoTset | 0.730 | 0.701 | 0.728 | 0.691 | 0.00031 |
| ANN | DNN-Edge-IIoTset | 0.980 | 0.983 | 0.980 | 0.981 | 0.00938 |

of 128 units/neurons (*i.e.,* 0 to 127). It takes the input from the first layer, applies a linear transformation, and then applies the ReLU activation function. This layer derives 128 distinct features from the input data. The third layer is also a fully connected layer abbreviated as Dense Layer 2 and consists of 64 units/neurons (*i.e.,* 0 to 63). It takes the input from the output of second layer, applies another linear transformation, and then applies the ReLU activation function. This layer derives 64 distinct features from the input data. The last layer is an output layer and consists of 15 units/neurons, corresponding to the 15 output classes. It takes the input from the output of third layer and transforms it into a 15 dimensional vector, where each element represents the probability of the input belonging to a particular class.

The findings and results of this study investigation provide critical insights into the effectiveness of the models used to detect and classify malicious IoT data. Importantly, the results conclusively indicate that the ANN performed better compared to the naïve Bayes and Decision Tree classifiers, with the exclusion of latency. This could be attributed to the powerful capabilities of ANNs as they can identify non-linear relationships in the data, extract relevant features from raw data by itself, and manage large datasets effectively. Moreover, the public available dataset Edge-IIoTset played an important role as a highly usable and feasible data source for researchers, and industries to navigate the difficulty caused by infectious IoT data flow.

The Figs. 4, 5 and 6 represent precision, recall, and F1-score of the selected three classifiers, calculated for ML-Edge-IIoTset, while the Figs. 7, 8 and 9 represent precision, recall, and F1-score of the selected three classifiers, calculated for DNN-Edge-IIoTset, respectively. The following figures encompass 15 distinct IoT data traffic classes pertaining to cybersecurity, represented on the *x*-axis. The *y*-axis represents the precision (in dark blue), recall (in gray), and F1-score scores (in light blue).

The performance of naïve Bayes in classifying various classes tends to vary, as depicted in Fig. 4. Naïve Bayes exhibits commendable performance in accurately classifying various types of data, such as *DDoS_UDP* with a high recall and F1-score and *Normal* with a low recall and F1-score. *DDoS_ICMP* recall and precision metrics are satisfactory, however, the F1-score exhibits a comparatively insignificant value, indicating the possible existence of an asymmetry between recall and precision. Additionally, *DDoS_TCP* and *Vulnerability_Scanner* are two classes that demonstrate a moderate level of performance. The metrics for the *Port_Scanning* class are zero, indicating that the model lacks the

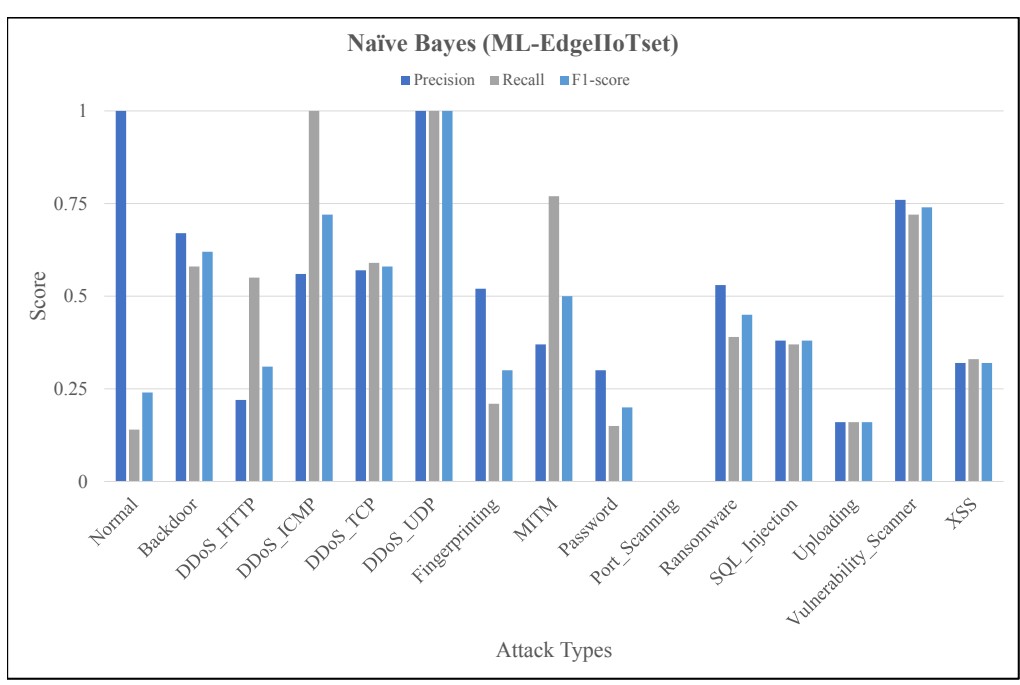

**Figure 4** Evaluation metrics for Naïve Bayes (ML-EdgeIIoTset).

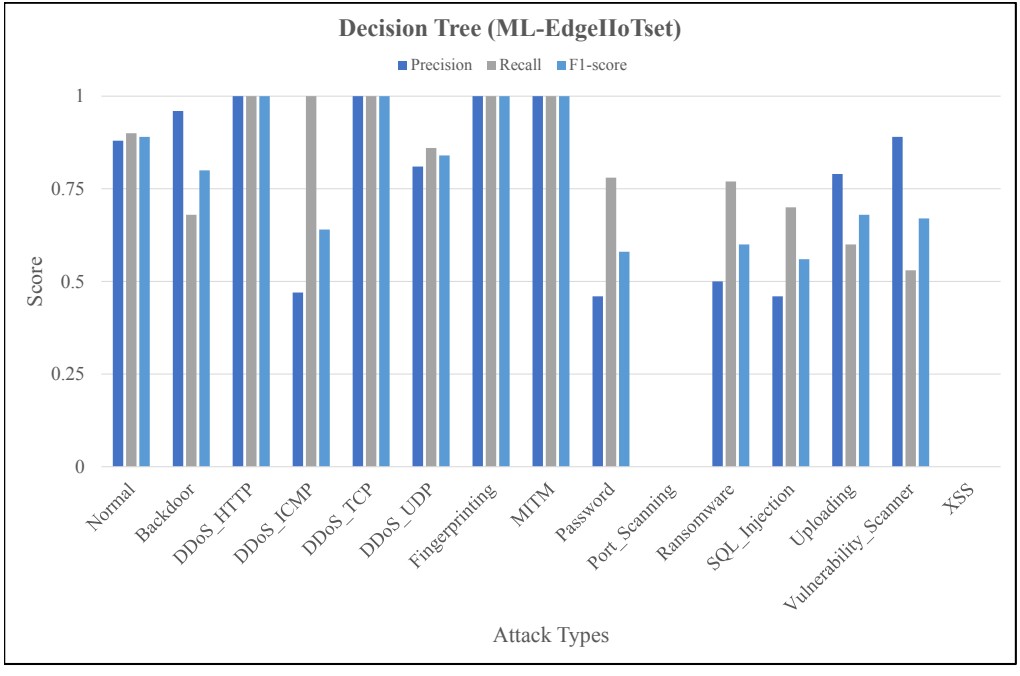

**Figure 5** Evaluation metrics for decision tree (ML-EdgeIIoTset).

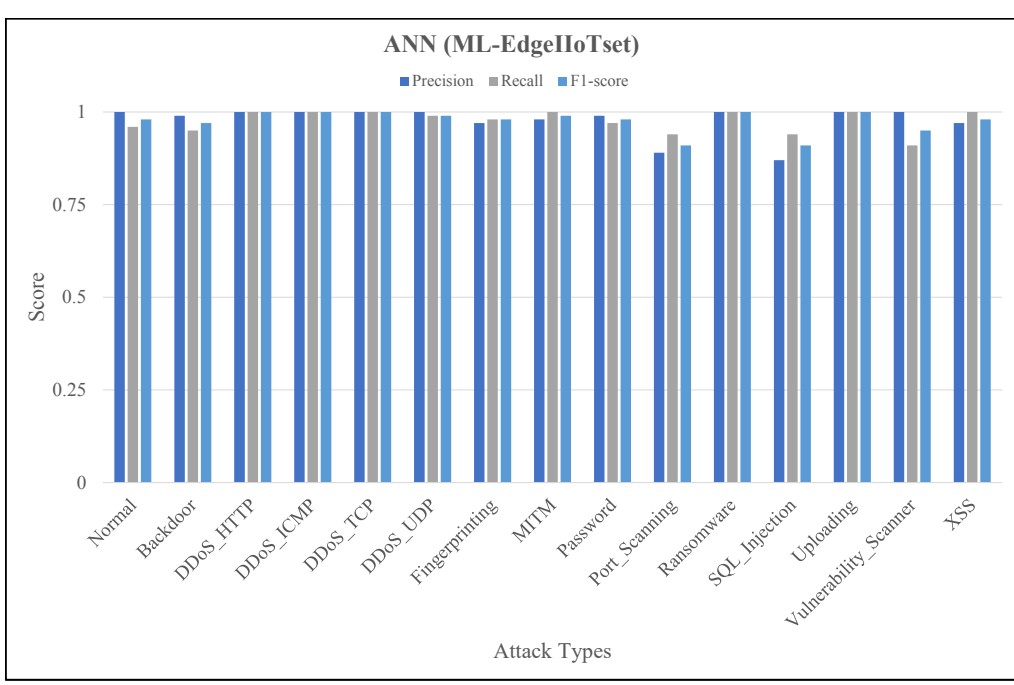

**Figure 6** Evaluation metrics for artificial neural network (ML-EdgeIIoTset).

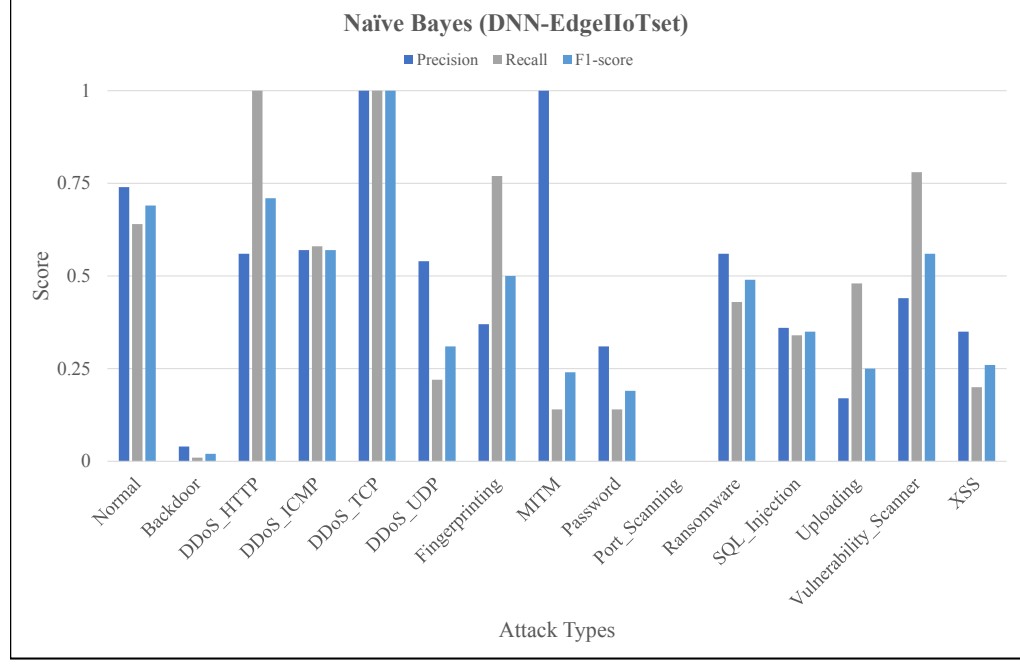

**Figure 7** Evaluation metrics for naïve Bayes (DNN-EdgeIIoTset).

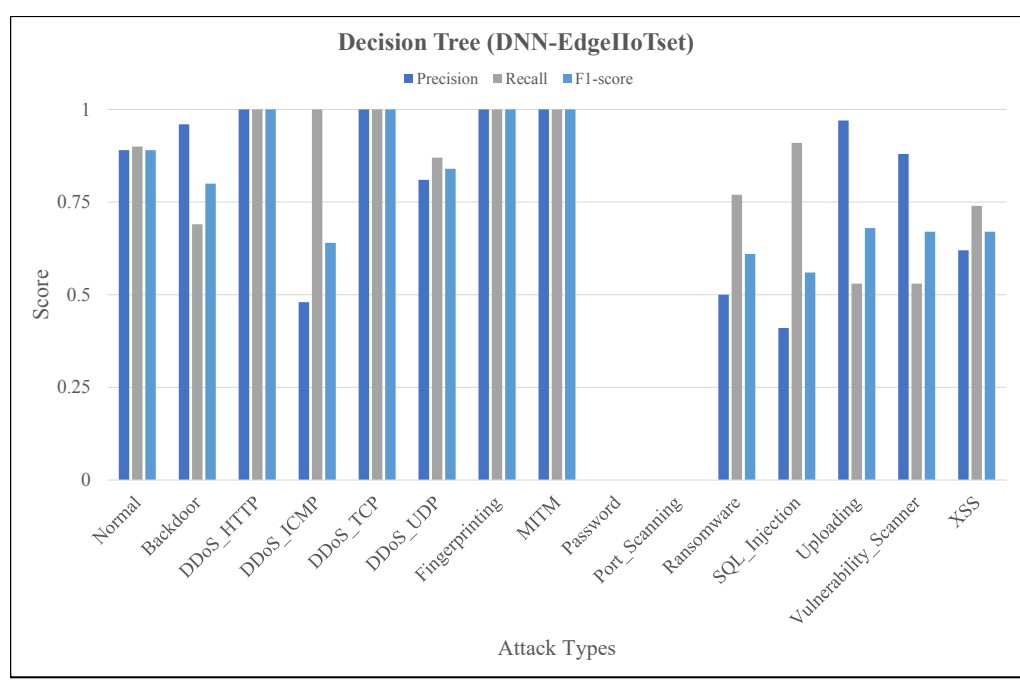

**Figure 8** Evaluation metrics for Decision Tree (DNN-EdgeIIoTset).

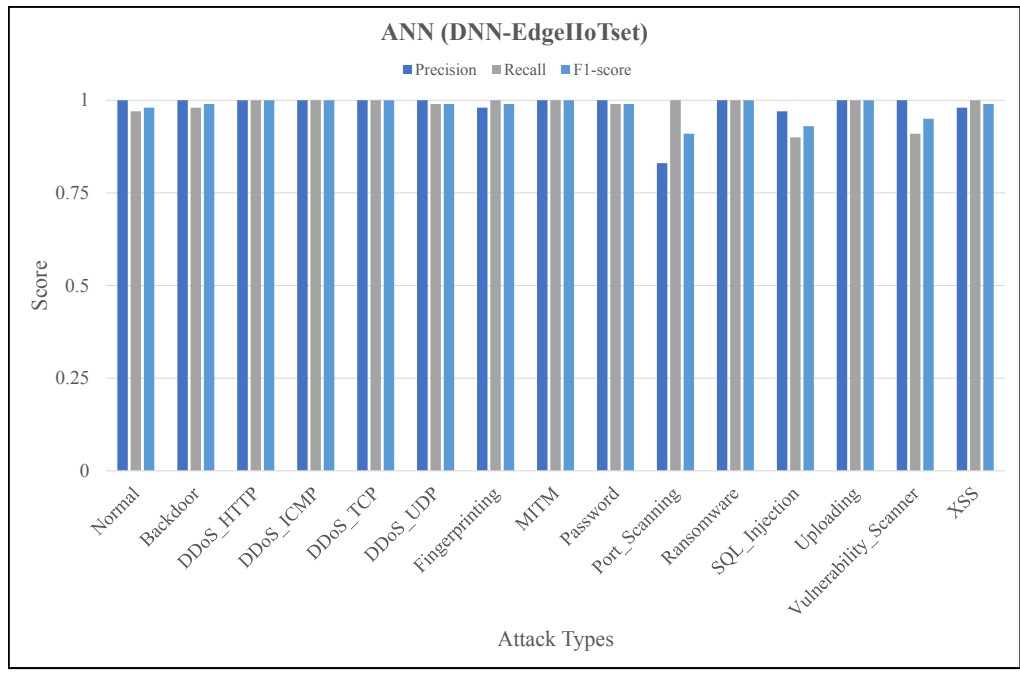

**Figure 9** Evaluation metrics for artificial neural network (DNN-EdgeIIoTset).

capability to precisely identify instances that belong to these classes. Figure 5 depicting evaluations of Decision Tree, demonstrates substantial improvements in multiple categories when compared to Fig. 4. However, certain classes, including *XSS* and *Port_Scanning* demonstrate minimal or nonexistent metrics, indicating that the model lacks the capability to precisely identify instances that belong to these classes. Figure 6 depicting evaluations of ANN, exhibits outstanding performance in the majority of classes on a consistent basis. Outstanding precision, recall, and F1-scores are demonstrated by the *Normal*, *Backdoor*, *DDoS_HTTP*, *DDoS_ICMP*, *DDoS_TCP*, *DDoS_UDP*, *Fingerprinting*, *MITM*, *Password*, *Ransomware*, *Uploading* and *XSS* classes. However, despite being slightly lower, the ratings for *Port_Scanning* and *SQL_Injection* remain within acceptable ranges. In summary, ANN consistently surpasses in terms of performance, naïve Bayes and Decision Tree in nearly all categories for ML-EdgeIIoTset.

Like Fig. 4, Fig. 7 shows variation in the performance of naïve Bayes across different classes. For instance, *DDoS_TCP* has perfect precision, recall, and F1-score, indicating a flawless classification for this class. However, classes like *DDoS_HTTP*, *Fingerprinting*, and *Vulnerability_Scanner* have relatively balanced scores, with moderate to high values for precision, recall, and F1-score. On the contrary, classes like *Backdoor* and *Port_Scanning* have very low scores across all metrics, indicating poor performance in identifying these classes. Figure 8 depicting evaluations of Decision Tree, demonstrates substantial improvements as compared to Fig. 7, notably in classes like *Backdoor*, *DDoS_HTTP*, and *Fingerprinting* where precision, recall, and F1-score have increased significantly. However, classes like *Password* and *Port_Scanning* still have poor performance, with precision, recall, and F1-score all being zero, indicating that the model fails to identify these classes at all. For ANN, in Fig. 9, there is a significant improvement across the board, with almost all classes achieving perfect or near-perfect scores in precision, recall, and F1-score. In summary, ANN surpasses in terms of performance, naïve Bayes and Decision Tree in all categories for DNN-EdgeIIoTset.

It can be seen in Table 4, that in terms of IoT attack detection, ANN demonstrates superior performance compared to both naïve Bayes and Decision Tree; however, with regard to latency, Decision Tree surpasses both naïve Bayes and ANN models. Therefore, it is not possible to designate any of these techniques as the optimal AI method, given that their performance varies across domains. To mitigate this issue, the proposed state-of-the-art evaluation metric LAAI is utilized to calculate a solitary outcome that comprehensively represents both performance and latency (see Table 4). Let us consider a hypothetical situation wherein AI Model-A attains an accuracy rate of 90% with a detection latency of 1 ms, whereas AI Model-B attains an accuracy rate of 75% with a detection latency of 0.1 ms. It is not feasible to determine an optimal model solely based on accuracy or latency in IoT domain due to the significance of both metrics. The proposed LAAI is preferable since it integrates both Accuracy and Latency. In the provided situation, the LAAI value for Model A is estimated to be 0.45, whereas, for Model B it is estimated to be 0.68. Therefore,

**Table 4  Proposed evaluation metric: latency aware accuracy index (LAAI).**

| AI model | Dataset | Accuracy | Latency (ms) | LAAI |
|---|---|---|---|---|
| Naïve Bayes | ML-Edge-IIoTset | 0.460 | 0.00030 | 0.459 |
| Decision tree | ML-Edge-IIoTset | 0.720 | 0.00011 | 0.719 |
| ANN | ML-Edge-IIoTset | 0.975 | 0.00765 | 0.967 |
| Naïve Bayes | DNN-Edge-IIoTset | 0.450 | 0.00046 | 0.449 |
| Decision tree | DNN-Edge-IIoTset | 0.730 | 0.00031 | 0.729 |
| ANN | DNN-Edge-IIoTset | 0.980 | 0.00938 | 0.971 |

**Table 5  ML-Edget-IIoTset sub datasets.**

| Dataset | Records |
|---|---|
| ML-Edget-IIoTset sample 1 | 52,600 |
| ML-Edget-IIoTset sample 2 | 52,600 |
| ML-Edget-IIoTset sample 3 | 52,600 |

**Table 6  DNN-Edget-IIoTset sub datasets.**

| Dataset | Records |
|---|---|
| DNN-Edget-IIoTset sample 1 | 739,733 |
| DNN-Edget-IIoTset sample 2 | 739,733 |
| DNN-Edget-IIoTset sample 3 | 739,733 |

confidently asserting that Model B is the superior choice for selection when compared to Model A in terms of accuracy and speed.

Latency is a critical element for IoT security because of resource limitations in IoT devices. In light of both performance and latency, the results obtained from employing LAAI indisputably indicate that the ANN exhibits superior performance in comparison to both the naïve Bayes and Decision Tree classifiers. To validate the accuracy and reliability of the acquired results (see Table 4), the datasets ML-Edge-IIoTset and DNN-Edge-IIoTset are each partitioned into three sub-datasets (see Tables 5 and 6). These sub datasets act as six distinct datasets with distinct data records, and are used to evaluate the performance of three selected classifiers/models (*i.e.,* naïve Bayes, Decision Tree and ANN) (see Tables 7 and 8).

Tables 7 and 8 show the extensive testing results and depicts the segmentation of the both datasets into three respective subsets of distinct records, which are used to assess the performance of the chosen models. It is evident that when it comes to detecting IoT attacks, ANN outperforms both naïve Bayes and Decision Tree. However, in terms of latency, Decision Tree performs better than both naïve Bayes and ANN models. The proposed LAAI evaluation metric asserts that ANN are the most advantageous choice in terms of both accuracy and latency. The obtained outcomes demonstrate that the results obtained from segmenting both datasets are nearly identical to the results obtained from the original ML-Edge-IIoTset and DNN-Edge-IIoTset. Hence, the results obtained from the original datasets are being used as a benchmark for comparing the results of the segmented

**Table 7  Evaluation metrics with latency for samples of ML-Edge-IIoTset.**

| AI model | Samples | Accuracy | Latency (ms) | LAAI |
|---|---|---|---|---|
| Naïve Bayes | Sample 1 | 0.460 | 0.00030 | 0.459 |
| Naïve Bayes | Sample 2 | 0.460 | 0.00029 | 0.459 |
| Naïve Bayes | Sample 3 | 0.460 | 0.00032 | 0.459 |
| | Mean | 0.460 | 0.00030 | 0.459 |
| Decision tree | Sample 1 | 0.720 | 0.00011 | 0.719 |
| Decision tree | Sample 2 | 0.724 | 0.00014 | 0.723 |
| Decision tree | Sample 3 | 0.716 | 0.00013 | 0.715 |
| | Mean | 0.720 | 0.00012 | 0.719 |
| ANN | Sample 1 | 0.972 | 0.00762 | 0.964 |
| ANN | Sample 2 | 0.976 | 0.00768 | 0.968 |
| ANN | Sample 3 | 0.978 | 0.00766 | 0.970 |
| | Mean | 0.975 | 0.00765 | 0.967 |

**Table 8  Evaluation metrics with latency for samples of DNN-Edge-IIoTset.**

| AI model | Samples | Accuracy | Latency (ms) | LAAI |
|---|---|---|---|---|
| Naïve Bayes | Sample 1 | 0.449 | 0.00050 | 0.448 |
| Naïve Bayes | Sample 2 | 0.448 | 0.00048 | 0.447 |
| Naïve Bayes | Sample 3 | 0.453 | 0.00041 | 0.452 |
| | Mean | 0.450 | 0.00046 | 0.449 |
| Decision tree | Sample 1 | 0.730 | 0.00034 | 0.729 |
| Decision tree | Sample 2 | 0.710 | 0.00030 | 0.709 |
| Decision tree | Sample 3 | 0.750 | 0.00029 | 0.749 |
| | Mean | 0.730 | 0.00031 | 0.729 |
| ANN | Sample 1 | 0.984 | 0.00934 | 0.974 |
| ANN | Sample 2 | 0.982 | 0.00933 | 0.972 |
| ANN | Sample 3 | 0.979 | 0.00929 | 0.969 |
| | Mean | 0.981 | 0.00934 | 0.972 |

datasets. These results additionally serve as a foundation for verifying the reliability of the achieved LAAI.

## CONCLUSIONS

This research primarily focuses on the importance and role of latency, which holds significant importance among various other issues specifically in IoT devices, including applications in smart agriculture. This study investigates two machine learning classifiers and one deep learning model on two subsets of the EdgeIIoTset dataset, with varying record volumes between the subsets. The ML-Edge-IIoTset contains a smaller number of records compared to the DNN-Edge-IIoTset. It is clear from the results that naïve Bayes works better for ML-Edge-IIoTset than for DNN-Edge-IIoTset, while Decision Tree and ANN work better for DNN-Edge-IIoTset. In this study, ANN technique demonstrated superior accuracy compared to the other two methods, namely naïve Bayes and Decision Trees.

However, Decision Tree exhibited superior speed as compared to all selected techniques. In order to determine the best appropriate method that achieves a balance between speed and performance, a novel approach called the Latency Accuracy Assessment Index (LAAI) was introduced. The analysis utilizing LAAI indicates that ANN consistently performs better than both naive Bayes and Decision Trees in any given instance. Hence, when dealing with IoT classification tasks, especially in the field of agriculture, it is crucial to extensively evaluate both latency and accuracy metrics. ANN is showing promise as a suitable option according to the suggested evaluation metric by achieving an optimal LAAI of 0.971.

### Funding
The authors received no funding for this work.

### Competing Interests
The authors declare there are no competing interests.

### Author Contributions
- Omar Bin Samin conceived and designed the experiments, performed the experiments, performed the computation work, prepared figures and/or tables, authored or reviewed drafts of the article, and approved the final draft.
- Nasir Ahmed Abdulkhader Algeelani performed the experiments, prepared figures and/or tables, authored or reviewed drafts of the article, and approved the final draft.
- Ammar Bathich analyzed the data, prepared figures and/or tables, and approved the final draft.
- Maryam Omar performed the experiments, performed the computation work, authored or reviewed drafts of the article, and approved the final draft.
- Musadaq Mansoor analyzed the data, authored or reviewed drafts of the article, and approved the final draft.
- Amir Khan conceived and designed the experiments, analyzed the data, authored or reviewed drafts of the article, and approved the final draft.

### Data Availability
The source code is available in the Supplemental Files.

The "Edge-IIoTset Cyber Security Dataset of IoT" (aka "EdgeIIoTset") dataset is available at Kaggle: https://www.kaggle.com/datasets/mohamedamineferrag/edgeiiotset-cyber-security-dataset-of-iot-iiot.

### Supplemental Information
Supplemental information for this article can be found online at http://dx.doi.org/10.7717/peerj-cs.2276#supplemental-information.

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
