# Peer review of "Optimizing agricultural data security: harnessing IoT and AI with Latency Aware Accuracy Index (LAAI)"

_PeerJ Computer Science, doi:10.7717/peerj-cs.2276_

## Round 0.1 · original submission · Major Revisions

The reviewers agreed that the paper is well-written and clear and that the topic is very relevant. However, the paper should be improved in some aspects before publication.

[1] The related work discussion misses some important approaches that should be added.
[2] The application context (agriculture) is used as motivation for the proposed approach, but it is only marginally considered in the paper. A stronger correlation between the application context and the proposed solution should be highlighted.
[3] The reviewers also have concerns about the validity of the results, as the empirical evaluation contains some ambiguities that should be cleared.
[4] In addition, the assumptions needed to corroborate the paper’s findings should be precisely stated, such as the underlying connectivity model considered.

Reviewer 1 ·

Basic reporting

It is a very relevant topic. The article is well-written and easy to follow. However, it needs some improvements.

Experimental design

1. ANN is a broad term. Please mention what structure you used to reach such high accuracy.
2. LAAI instead of latency is shadowing the latency term instead of normalization. The result should be discussed in the light of latency and accuracy. If the authors want to go ahead with LAAI, stronger documental support is a must.

Validity of the findings

LAAI instead of latency is shadowing the latency term instead of normalization. The result should be discussed in the light of latency and accuracy. If the authors want to go ahead with LAAI, stronger documental support is a must.

Reviewer 2 ·

Basic reporting

Overall the article is written well by the authors, however following are a few observations where the authors may improve:

1. The title of the article emphasizes on Optimizing agricultural data security but the article mostly talks about IIoT and AI/ML except a few lines on Agriculture in the Introduction section.

2. The English language used to write the article is satisfactory but it can be improved. In some of the instances the sentences are either incomplete or authors are unable to convey the meaning. E.g. line numbers: 143, 157, 158, 159, etc.

3. In the literature review section, the authors have tried to explore the literature (08 articles) related to AI/ML based approach for IoT based security (especially IDS). Though articles of other domains like healthcare, geoscience and multimedia have been included by authors, but they missed articles related to Agriculture IoT and its security. E.g.:
a) https://doi.org/10.1002/9781119769231.ch11
b) https://doi.org/10.1007/978-3-030-21952-9_2
c) https://doi.org/10.1016/j.inpa.2023.09.002
d) http://doi. org/10.1016/j.biosystemseng.2019.12.013
e) https://doi.org/10.3390/app12073396
f) https://doi.org/10.1016/j.compag.2021.106352

4. The concept of introducing Latency Aware Accuracy Index (LAAI) seems promising at a larger scale provided that a rigorous study is carried out and the results are supported with the diverse set of experiments. The concept of Latency + AI (E.g.: https://doi.org/10.1109/TAI.2024.3366880 and https://doi.org/10.3390/s22197326) has already been explored by other researchers and merely attaching it with Cyber Security problem pertaining to IoT and showcasing it with the Agriculture is not significant.

5. Figures 1 & 2 are very basic.

6. The Dataset section can be rewritten by keeping only important and relevant things as it currently occupies almost 10% of the article.

Experimental design

The authors presented experiments and results to introduce a novel approach to assist selection of an ML/DL model for security of IoT based agriculture. However, the experiments, results and aim do not match as there are a lot of ambiguities, some of them are listed below:

1. On line numbers 194 & 195 the authors mentioned that the ML dataset is suggested to be utilized with ML Algorithms and DL Dataset with DL algorithms. On lines 219 & 220 the authors classified NB and DT as ML models while ANN as DL model. But at the time of designing experiments the authors used all three algorithms NB, DT & ANN with both the datasets.

2. The authors introduced LAAI (line #248 to #250) claiming it to be a novel and comprehensive evaluation measure... and mentioned that Higher LAAI Score indicates better performance and lower score indicates low performance (lines #257 to #260). The derivation of these statements are proved based on simple experiments performed on EdgeIIoT dataset and hence lack the benchmarking of results. Additionally, as per the formula and as the results discussed in the article the LAAI is very close to Accuracy which raises a concern. The authors may test LAAI on a diverse set of datasets to make the claims.

Validity of the findings

The results discussed by the authors are based on two sets of experiments (each with 03 models NB, DT and ANN) performed on an existing dataset. The authors tried to discuss the importance of LAAI (introduced by them) but the results are not sufficient enough to reach any conclusion.

Reviewer 3 ·

Basic reporting

The authors propose a novel evaluation metric titled 'Latency Aware Accuracy Index (LAAI)' for the purpose of optimising data security in the agricultural sector.
The topic discussed in this article is absolutely worthy of attention and of great interest to those who develop technologies aimed at solving concrete problems in agricultural production processes.

The literature is focused solely and exclusively on methods for presiding over security in the IoT field. Recent references should be added on this issue when it is applied to the domain of agriculture.

The proposal aims at defining an AI-based method to classify traffic to and from ioT nodes in order to intercept malicious intrusions. How does this approach compare to other models that eliminate the problem at the root? I am thinking for example of the use of block-chain to guarantee the integrity of traffic. Are these alternative or complementary methods? Why?

Since latency factors are important for this paper, they should already be defined in the introduction. A summary of the concept and its implications in this context makes the paper more readable.
170 to 193 list the classes of attacks in the Edge-IIoTset dataset. It would be useful to report for each class also some concrete examples of the effects they could have in agriculture, also indicating a severity scale of the attacks for the effects they would have (e.g. during plant growth the manipulation of a data read by an NPK sensor could have very serious consequences in the supply of agronomic inputs with direct effects on production costs, soil pollution and nitrate accumulation in the final products). The general observation is that the article deals marginally with the application context.
In 207 what criteria were used in order to minimise the generation of unrealistic data. Please explain.
Conclusion
The performance of an iot system applied to agriculture depends heavily on the underlying connectivity model (e.g. request-response, publish-subscribe, push-pull, exclusive pair). These models are not included in the chosen dataset, could they influence the results obtained? I recommend arguing this aspect in the conclusions as well.

Experimental design

no comment

Validity of the findings

no comment

Additional comments

no comment

---

## Round 0.2 · accepted · Accept

I carefully read the revision and the authors’ response to reviewers. The authors addressed all reviewers’ comments and improved the overall presentation of the paper. The paper is ready for publication.